# A Novel Exercise for Enhancing Visuospatial Ability in Older Adults with Frailty: Development, Feasibility, and Effectiveness

**DOI:** 10.3390/geriatrics5020029

**Published:** 2020-05-03

**Authors:** Miyuki Nemoto, Hiroyuki Sasai, Noriko Yabushita, Keito Tsuchiya, Kazushi Hotta, Yoshihiko Fujita, Taeho Kim, Takehiko Tsujimoto, Tetsuaki Arai, Kiyoji Tanaka

**Affiliations:** 1Dementia Medical Center, University of Tsukuba Hospital, Tsukuba 3058575, Japan; 2Research Team for Promoting Independence and Mental Health, Tokyo Metropolitan Institute of Gerontology, Tokyo 1730015, Japan; hiroyuki.sasai@gmail.com; 3Upten Health Support, Tsukuba 3050047, Japan; yabushita-noriko@upten.jp; 4School of Health and Physical Education, University of Tsukuba, Tsukuba 3058577, Japan; keito19920904@gmail.com; 5Department of Occupational Therapy, Ibaraki Prefectural University of Health Sciences, Ami 3000394, Japan; hotta@ipu.ac.jp (K.H.); yoshihiko.fujita@gmail.com (Y.F.); 6Graduate School of Comprehensive Human Sciences, University of Tsukuba, Tsukuba 3058577, Japan; kth3442@gmail.com; 7Faculty of Human Sciences, Shimane University, Matsue 6900823, Japan; tsujimoto@hmn.shimane-u.ac.jp; 8Faculty of Medicine, University of Tsukuba, Tsukuba 3058575, Japan; 4632tetsu@md.tsukuba.ac.jp; 9Faculty of Health and Sport Sciences, University of Tsukuba, Tsukuba 3058577, Japan; tanaka.kiyoji.ft@u.tsukuba.ac.jp

**Keywords:** cube exercise, visuospatial ability, frailty

## Abstract

We aimed to develop a novel exercise to improve visuospatial ability and evaluate its feasibility and effectiveness in older adults with frailty. A non-randomized preliminary trial was conducted between June 2014 and March 2015. We recruited 35 adults with frailty (24 women), aged 66–92 years. Participants were assigned to either locomotive- or visuospatial-exercise groups. All participants exercised under the supervision of physiotherapists for 90 min/week for 12 weeks. The visuospatial exercise participants used cubes with six colored patterns and were instructed to “reproduce the same colored pattern as shown in the photo”, using the cubes. In the locomotive exercise group, lower extremity functional training was provided. Rates of retention and attendance measured feasibility. Most participants completed the intervention (77.3%, locomotive; 84.6%, visuospatial) and had good attendance (83.8%, locomotive; 90.7%, visuospatial). Mini-mental state examination (MMSE), clock drawing test (CDT), and seven physical performance tests were conducted before and after interventions. The improvement in the MMSE score, qualitative analysis of CDT, grip strength, and sit and reach assessments were significantly greater in the visuospatial exercise group than in the locomotive exercise group. The cube exercise might be a feasible exercise program to potentially improve visuospatial ability and global cognition in older adults with frailty.

## 1. Introduction

Frailty is a well-recognized geriatric syndrome that should be managed to prevent older adults from becoming increasingly dependent [1]. Frailty is associated with cognitive impairment [2], and older adults with physical frailty have a 1.3 to 2.7 times greater risk for developing dementia than those without it [3]. Therefore, managing frailty at early stages is important to delay functional and cognitive declines [4].

Several studies reported that the risk of developing dementia, irrespective of its sub-categories including Alzheimer’s disease (AD), mild cognitive impairment (MCI), and cognitive decline, could be significantly reduced with exercise [5]. However, little is known about the effectiveness of exercise interventions on different domains of cognition, such as attention, memory, executive function, and visuospatial ability in older adults with frailty [4].

Visuospatial function is the ability with which a person specifies the configuration of an object, analyzes its position in space, integrates a coherent spatial framework, and performs mental operations on spatial concepts [6]. This ability is impaired at the early stages of AD [7], and is predictive of future falling, getting lost, and apraxia. It is also considered an essential factor to maintain activities of daily living (ADL) [8].

To improve visuospatial ability, computerized cognitive training has been extensively evaluated and some training has been found to improve visuospatial function [9]. Yet, several challenges to providing this training, such as the difficulty of accessing some pieces of equipment and limited training areas, limit its widespread application in clinical practice. Therefore, we developed a novel visuospatial exercise program (an exercise program combined with cognitive tasks) named cube exercise. In this cube exercise, an image of a three-dimensional structure provides stimuli to the occipital–parietal and temporal lobes, potentially enhancing visuospatial ability. In addition, the set of blocks (i.e., cubes) is light and portable, and is easy to use anywhere. Our newly developed cube exercise may have advantages in simultaneously enhancing both intellectual activity (arranging cubes) and physical activity (elements of exercise). These two elements are thought to be preventive factors of cognitive impairment.

Therefore, the purpose of this study was to develop and evaluate the feasibility of “cube exercise” among older adults with frailty. In addition, we examined whether cube exercise could improve visuospatial ability in older adults with frailty.

## 2. Materials and Methods

### 2.1. Design and Setting

This, 12-week, two-arm, non-randomized, and controlled trial was conducted at a public health center in Yachiyo-town, Ibaraki, Japan, from June 2014 to March 2015. The participants, who were recruited in 2014, were assigned to the locomotive exercise group, and those in 2015 were assigned to the visuospatial exercise group. The intervention was physical exercise in both groups. Since the locomotive training was routinely provided as part of the local government health services, we considered the locomotive training as a control intervention. The cube exercise was provided on a trial basis. Considering the constraints of location, time, and human resources as a collaborative project with the town, a non-randomized study design was adopted. This study was conducted in accordance with the Declaration of Helsinki, and the study protocol was reviewed and approved by the Ethics Committee of the Faculty of Health and Sport Sciences, University of Tsukuba, Japan (approval number: 25–96). All participants provided written informed consent. This manuscript is fully complied with the TREND statement [10] which is a well-recognized reporting guideline for non-randomized trial.

### 2.2. Participants

We recruited older adults with frailty according to the Long-Term Care Insurance system in Japan [11] by distributing the Kihon Checklist (KCL) [12] to adults aged 65 years or older in Yachiyo town. The KCL has been validated both nationally and internationally to identify older adults with frailty [13,14]. The KCL consisted of 25 dichotomous questions within six domains: activities of daily living (five items), physical function (five items), nutritional status (two items), oral function (three items), cognitive function (five items), and depressive mood (five items). Town’s public health nurses encouraged older adults who met ≥3/5 items in the physical function domain to participate in this trial. The other inclusion criteria were individuals able to walk regardless of using a walking aid. Exclusion criteria were (i) participants with inability to understand the instructions and perform physical performance tests, (ii) presence of terminal disease or progressive deterioration of health, (iii) presence of history of any neurological disease (e.g., physician-diagnosed dementia, stroke and Parkinson’s disease) with residual impairment, and (iv) being judged as ineligible according to the manual of the Japanese Ministry of Health, Labor and Welfare on exercise intervention for older adults with frailty [15] by the principal investigator. (K.T.). The manual includes exclusion criteria for participation in exercise programs. For example, medical judgment recommends the exclusion of patients with circulatory and the musculoskeletal disorders, which have a high risk of worsening disease with exercise.

### 2.3. Program Development

Cube exercise was inspired by children playing a game of building blocks. We thought that assembling blocks with matching patterns could have a beneficial effect on cognitive function, especially visuospatial function. Visuospatial function is commonly conceptualized in three components: visual perception, construction, and visual memory [16]. The task of recognizing figures and colors, and combining cubes could train contrast sensitivity, which involves visual perception and composition ability. If playing with building blocks could use the whole body, then cube exercise could be a good program for both physical and cognitive functions. To make building blocks a dynamic exercise, we made the cubes bigger and heavier than the building blocks. We determined an appropriate size and weight for the cube so that older people can exercise safely. The program was developed to make it possible to effectively perform the exercises for enhancing cognitive and physical functions. The degree of difficulty and program variations were provided by physiotherapists.

### 2.4. Interventions

The participants in both locomotive and visuospatial groups exercised under the supervision of physiotherapists for 90 min/day, 1 day/week, for a 12-week period. The programs included a 10-min warm-up, followed by 20 min of muscle strengthening, and 20 min of nutrition lecturing. For the remaining 40 min, the participants engaged in either locomotive or visuospatial exercise according to their group allocation. All exercises were performed at a moderate intensity (rate of perceived exertion [17] of 11–13) for both groups.

#### 2.4.1. Locomotive Exercise

Participants in the locomotive exercise group were taught the correct way of walking, obstacle walking, and other exercises as a group exercise. Locomotive exercise aimed to maintain and improve lower extremity functions. The exercise consisted of a 10 m walk in weeks 1–4, for which physiotherapists instructed the participants in the correct way of walking; adding obstacles to the 10 m course in weeks 5–8, in which participants avoided the obstacles; and adding a 5 m ladder to the walking course in weeks 9–12, in which participants walked regularly, sideways, and crossed using the ladder.

#### 2.4.2. Visuospatial Exercise

The visuospatial exercise program was named “cube exercise”. Cube exercise aims to enhance visuospatial cognitive function. This exercise includes a dual task element that uses the brain while moving the body. We used colored cubes (30 × 30 × 30 cm, 480 g, made of styrofoam), with six patterns using blue, white, yellow, and pink for each surface. Participants assembled the blocks according to a model picture. They were asked to “reproduce the same colored pattern as shown in the model picture, using a set of colored cubes”. The 12-week program had three levels, in which the difficulty gradually increased; in weeks 1–4, participants assembled in only one dimension (level 1); in weeks 5–8, participants assembled in two dimensions (level 2); in weeks 9–12, participants assembled in three dimensions (level 3, Figure 1). Cube exercise could be assembled by one, two, or more persons. Cube exercise was performed as shown in Figure 2. Two teams played against each other. The cubes and the model picture were placed 10 m ahead, then the participants walked to that point and assembled the cubes. When a participant finished assembling the cubes, the instructor checked the structure. If correct, the participant returned to his/her team and was replaced by the next person. The team that correctly assembled all blocks first won. The exercise comprised cognitive training and physical activity. Participants tried to imagine a three-dimensional structure, and then they lifted, carried, and placed the cubes into the correct formation.

### 2.5. Feasibility Assessment

We evaluated feasibility by retention rate (proportion of participants who completed the post-intervention assessment), attendance rate (the number of attendances/total number of sessions), and diary review. Participants were instructed to record daily exercise and program impressions in a printed diary that we had prepared. Physiotherapists reviewed each diary to assess the participant’s impression of cube exercise.

### 2.6. Preliminary Effectiveness Assessment

All outcome measures were assessed before and after the 12-week intervention by trained staff members, including physiotherapists. The primary outcomes comprised changes in the mini-mental state examination (MMSE) and qualitative and quantitative analyses of the clock drawing test (CDT). The exploratory outcomes were changes in various physical function tests.

*Anthropometric variables:* Height and weight were measured in light clothing without shoes, and body mass index (BMI, kg/m^2^) was calculated.

*Self-reported measures of physical function:* Functional status was evaluated through instrumental ADL [18]. This scale consists of eight items (0–8 points): using telephone, shopping, preparing meals, housework, laundry, using transportation, managing medication, and managing property. A higher score indicates independence. The physiotherapist administered the test.

*Cognitive function:* The MMSE and CDT were assessed by speech therapists. MMSE is designed to quickly measure global cognitive functioning, temporal and spatial orientation, attention, immediate and short-term memory, language, praxis, and calculation. Scores ranged from 0 to 30, with higher scores indicating better cognitive performance [19].

CDT is a method to evaluate visuospatial cognitive function. In recent years, it has been used as a neuropsychological screening for cognitive impairment and dementia, especially for the initial stage of AD [20]. CDT consists of understanding, planning, visual memory and recalibration of graphic images, visual space function and execution, recognition of numbers, abstraction, and concentration [21]. Participants are asked to draw a clock with the time reading 10 past 10, providing a measure of visual memory. We evaluated CDT by pre-planned qualitative and quantitative analyses using the Freedman method [22] (Appendix A) and Rouleau’s method [23]. Kaplan [24] reports that qualitative analysis is more important than quantitative analysis for understanding our brain function. We classified the results into three categories: improved, declined, and unchanged in qualitative analysis. In addition, in the qualitative evaluation of CDT, there were several typical errors (Appendix A). We categorized these errors using qualitative analysis: [25] “conceptual deficit” for a, b, and c; “perseveration” for d; and “spatial and/ or planning deficit” for e and f.

*Physical function tests:* Trained examiners assessed the physical performance twice, and the averaged value was used in the analysis (Appendix A). We carried out seven physical function tests: grip strength, five chair sit-to-stand, one-leg stance, sit and reach, timed up and go, usual gait speed, and hand working with peg board [26,27,28,29,30,31,32].

### 2.7. Statistical Analysis

Values were expressed as the mean and standard deviation at baseline. The 12-week changes in all outcome measures were expressed as mean and 95% confidence intervals (CI), to better interpret significance for within-group changes. Chi-squared tests were used to compare CDT change for both qualitative and quantitative analyses at baseline and week 12 between the two groups. Repeated analysis of covariance was used to determine between-group differences in cognitive or physical function changes with adjustment for age, sex, educational attainment, and baseline values of cognitive function or physical function [33,34,35]. Missing data were replaced by baseline value. The data were analyzed using IBM SPSS Statistics, version 22 (IBM Corporation, Armonk, NY, USA) with a significance level of 5%.

## 3. Results

The participant flow is illustrated in Figure 3. A total of 22 candidates in the locomotive exercise group and 13 candidates in the visuospatial exercise group fulfilled the eligibility criteria and were enrolled. After 12 weeks, 17 (77.3%) of the locomotive exercise group and 11 (84.6%) of the visuospatial exercise group completed the post-intervention assessments. The reasons for missing assessments were mostly scheduling conflicts and medical reasons unrelated to this trial. The attendance rate was 83.8% (28.6%–100%) for the locomotive exercise group and 90.7% (64.3%–100%) for the visuospatial exercise group, with no significant difference in attendance between the two groups.

Based on diary review, positive opinions included ‘it was fun like a puzzle’, ‘It was a program that would work for both the brain and the body’, ‘It was nice to enjoy it alone or in a group’, and ‘It was good that the difficulty gradually increased’. Negative opinions included ‘It was difficult to assemble cube according to the model photo’, and ‘I was tired with my brain’. Women tended to be slightly more alert than men. The program ended without injury or other adverse events.

Table 1 shows the baseline characteristics. This study included 35 older adults with frailty (24 women), aged 66–92 years (80.1 [7.3] years). Weight and BMI were significantly smaller in the locomotive exercise group than in the visuospatial exercise group. More participants had low-back pain and knee pain in the visuospatial exercise group than in the locomotive exercise group. Cognitive functions were similar between the two groups. The MMSE score ranged from 22 to 30 points at baseline. Among the physical functions, usual gait speed was faster in the locomotive exercise group than the visuospatial exercise group (*p* < 0.01) at baseline.

Table 2 shows the 12-week changes in cognitive and physical functions in the two groups. The improvement in MMSE score was significantly greater in the visuospatial exercise group than in the locomotive exercise group. We examined between-group differences in CDT scores. No significant between-group difference in 12-week CDT change was observed. Among the physical function tests, grip strength and sit and reach were improved significantly in the visuospatial exercise group as compared to the locomotive exercise group.

A graphical representation of qualitative CDT assessment is described in Figure 4. We classified the results into three categories: improved, declined, and unchanged. Five participants (22.7%) improved, five (22.7%) declined, and twelve (54.5%) were unchanged using the qualitative evaluation of CDT in the locomotive exercise group. Similarly, eight (61.5%) improved, none (0%) declined, and five (38.5%) showed no change in CDT in the visuospatial exercise group. The visuospatial exercise group showed significant improvement following the intervention (χ^2^: 6.70, *p* < 0.05).

## 4. Discussion

In the present study, we developed a novel exercise program called “cube exercise”, which we hypothesized would be a feasible program and might improve visuospatial ability. To the best of our knowledge, this is the first attempt to approach visuospatial ability training using exercise in older adults with frailty. The study demonstrated that cube exercise is a feasible and implementable exercise program for older adults with frailty. We also found that the cube exercise elicited greater improvements in MMSE score, grip strength, and sit and reach than locomotive exercise did. Furthermore, qualitatively assessed CDT tended to improve more in the visuospatial exercise group than in the locomotive exercise group. Although careful interpretation should be made by considering a non-randomized design of our study, these findings support that the cube exercise may have potential to improve cognitive and physical functions in older adults with frailty compared to a routinely prescribed locomotive exercise.

In the visuospatial exercise group, the attendance and retention rates were very high at 90.7% and 84.6%, respectively. A cube exercise program could be continuously implemented with less mental and physical burdens on participants. As frailty affects both physical and mental status, it is often difficult to continue participating in exercise or other behavioral programs [36]. However, this study achieved high attendance and retention rates. Participants’ feedback on cube exercise was generally positive. Cube exercise had a wide variety of contents. Setting the difficulty of the program might motivate participants to continue the cube exercise program. In addition, a program that could be implemented individually or in a group might increase the enjoyment of participants. Cube exercise could be developed into various programs according to participants’ physical strength and cognitive level.

Qualitative analysis of CDT showed that the cube exercise program might have a beneficial impact on the participant’s visuospatial ability. Improvement in spatial and/or planning deficit was found in four out of five persons with improved CDT in the visuospatial exercise group. As spatial and/or planning deficit might be caused by the decline in cognitive and executive functions [37], the cube exercise might be effective in improving the visuospatial ability and executive function of these participants. In fact, the participants in the visuospatial exercise group improved in the placement of the numbers and the center of the clock after the 12-week intervention. Our results indicate that they were able to better visualize the space. 

The present study also found that cube exercise might have a beneficial effect on global function (MMSE score). Exercises with complex/dual task elements have a positive effect on cognitive function. A previous study reported that a dual-task cognitive intervention program improved memory function in MCI [38]. Suzuki et al. also reported that multi-component exercise was effective in maintaining the global function of MCI in a six-month exercise intervention study in 100 MCI participants [39]. Cube exercise, which contained dual-task stimulation with a high cognitive load, might have a greater benefit on various cognitive functions than interventions that only focus on locomotive exercise.

Grip strength and sit and reach were significantly increased in the visuospatial group. Lipardo et al. reported that exercise and cognitive training improve physical performance and cognitive function [40]. It is conceivable that the action of lifting, carrying, and placing a large cube improved the muscular strength and walking ability of the upper limbs as well. 

The present study has several limitations. First, this study was a non-randomized trial with a small sample size. Non-randomized design introduces confounding biases and may possibly alter findings of this trial. Although several well-established covariates such as age, sex, education, and baseline value of each outcome were statistically controlled, the sufficient adjustment was not achieved due to a limited sample size. Thus, the existence of residual confounding could not be denied. Second, this study did not adopt examiner and evaluator blinding. A lack of blinding may lead to biased assessment and evaluations of outcomes. Notably, one of the authors performed a qualitative analysis of the CDT. However, the qualitative analysis strictly adhered to the Freedman’s quantitative evaluation items and Rouleau’s criteria. Since this study was part of a community health project with limited budgetary and human resources, non-blinding was feasible and practical choice. We should tackle these challenges when designing future trials. Therefore, we admit that the effect size estimates obtained from this study included substantial biases. To obtain a less-biased estimate of the effect size, we first plan to conduct a small pilot randomized controlled trial (RCT), in which randomization and examiner and evaluator blinding are introduced. As the next step, we would like to design a large-scale RCT to test the effectiveness of this program on cognitive function. Third, it may be difficult for participants to continue the program at home by themselves. However, cube exercise has the advantage of being a program that can be enjoyed within a group. Exercising, while communicating, has a positive effect on cognitive function. Therefore, we should develop another form of cube exercise that can be performed daily at home or in non-supervised settings. 

## 5. Conclusions

Our findings suggest that the cube exercise is a feasible exercise program for older adults with frailty. Regarding effectiveness, the cube exercise may be beneficial not only for visuospatial ability and global cognitive function, but also for physical function in older Japanese adults with frailty. Further study is required to test the effectiveness of the visuospatial training on a larger scale using a randomized design.

## Figures and Tables

**Figure 1 geriatrics-05-00029-f001:**
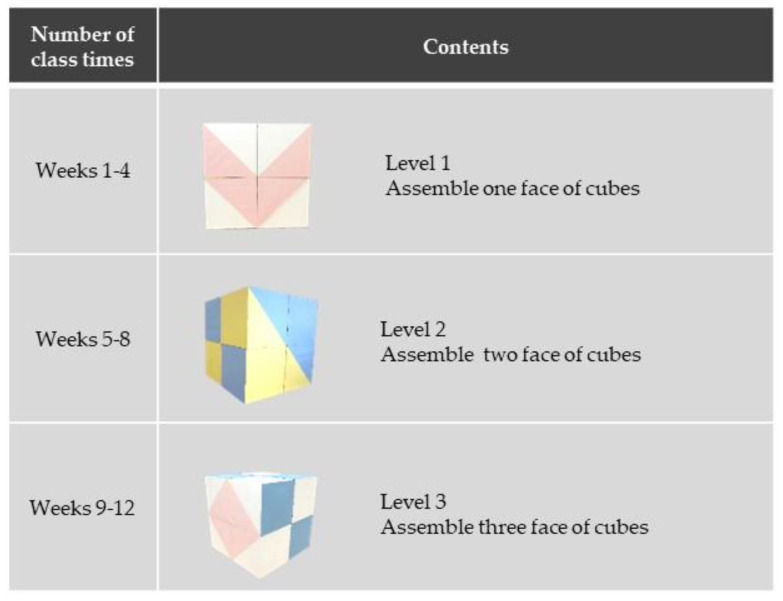
The contents of the cube exercise.

**Figure 2 geriatrics-05-00029-f002:**
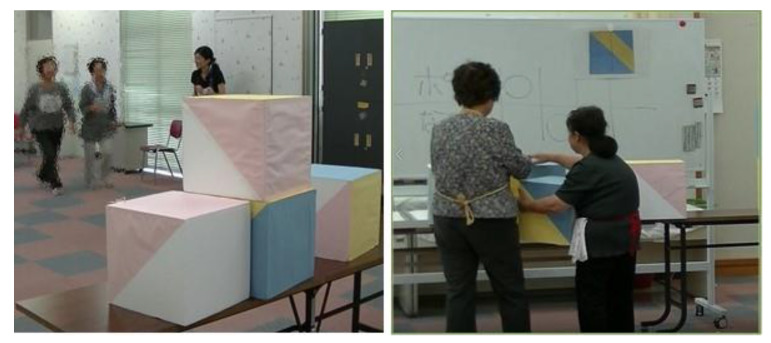
Practicing the cube exercise.

**Figure 3 geriatrics-05-00029-f003:**
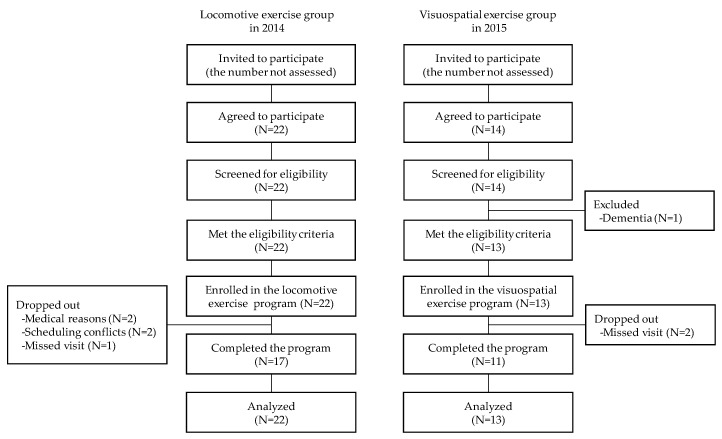
Flowchart of the study participants.

**Figure 4 geriatrics-05-00029-f004:**
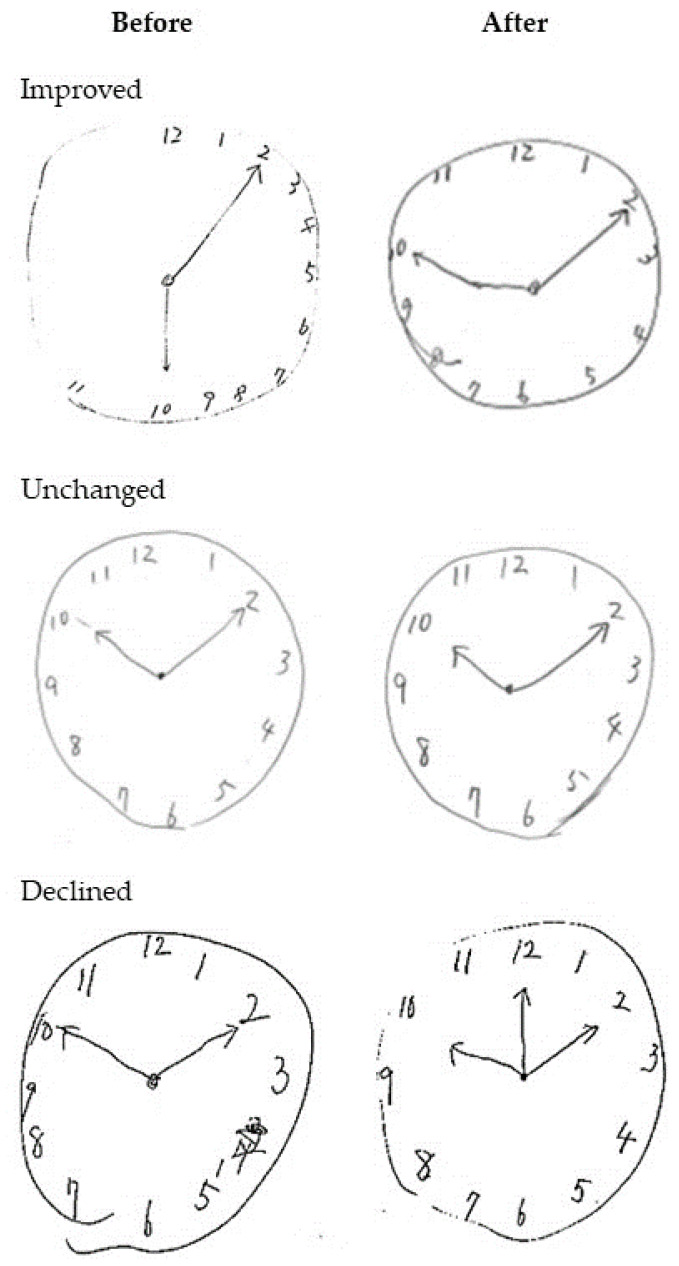
Examples of changes in Clock Drawing Test.

**Table 1 geriatrics-05-00029-t001:** Characteristics of the participants.

	Locomotive Exercise (N = 22)	Visuospatial Exercise (N = 13)	*p*-Values	Cohen’s d
**Demographics and anthropometrics**				
Sex (women), n (%)	17 (77.3)	7 (53.8)	0.26	0.24
Age, years	81.3 (5.9)	78.1 (9.1)	0.21	0.44
Height, cm	146.2 (9.3)	150.9 (9.9)	0.16	0.49
Weight, kg	48.5 (8.2)	57.6 (14.5)	0.02	0.83
Body mass index, kg/m^2^	22.7 (3.1)	25.1 (4.5)	0.07	0.65
Instrumental ADL scale, pts	6.9 (1.7)	7.3 (0.9)	0.45	0.27
**Medical history**				
Cerebrovascular disease, n (%)	3 (14.3)	2 (15.4)	1.00	0.02
Hypertension, n (%)	10 (45.5)	11 (84.6)	0.34	0.39
Diabetes mellitus, n (%)	4 (19.0)	2 (15.4)	1.00	0.05
Osteoporosis, n (%)	2 (9.5)	2 (15.4)	0.63	0.09
Low-back pain, n (%)	1 (4.8)	5 (38.5)	0.02	0.43
Knee pain, n (%)	0 (0.0)	3 (23.1)	0.05	0.40
**Cognitive functions**				
Mini-Mental State Examination, pts	25.3 (3.4)	25.3 (2.7)	0.97	0.00
Clock Drawing Test, pts	11.8 (3.1)	11.9 (2.3)	0.91	0.04
**Physical functions**				
Grip strength, kg	19.5 (5.0)	18.9 (6.3)	0.76	0.11
One-leg stance, s	11.5 (10.9)	8.3 (7.4)	0.38	0.33
Sit and reach, cm	29.8 (9.2)	26.7 (9.9)	0.38	0.33
Five chair sit-to-stands, s	10.8 (4.4)	11.4 (5.1)	0.74	0.13
Timed up and go, s	9.5 (2.2)	13.4 (8.8)	0.54	0.70
Usual gait speed, s	4.4 (1.4)	7.9 (3.8)	<0.01	1.37
Hand working with peg board, n	30.9 (7.7)	31.7 (9.4)	0.79	0.10

Data are shown as mean (standard deviation) for continuous variables and frequency (percentage) for categorical variables. ADL, activities of daily living.

**Table 2 geriatrics-05-00029-t002:** Changes in cognitive and physical functions following 12-week locomotive (control arm) or visuospatial (intervention arm) exercise programs.

Outcomes	Locomotive Exercise (N = 22)		Visuospatial Exercise (N = 13)		*p*-Values	Cohen’s d
**Cognitive functions ***						
Mini-Mental State Examination, pts	0.2	(−0.5, 1.0)	1.6	(0.6, 2.6)	0.04	0.79
Clock Drawing Test, pts	0.2	(−0.6, 1.1)	1.4	(0.3, 2.5)	0.09	0.70
**Physical functions ****						
Grip strength, kg	−0.1	(−1.1, 0.9)	2.0	(0.7, 3.3)	0.02	1.07
One-leg stance, s	2.2	(−3.0, 7.3)	2.2	(−4.6, 9.0)	0.99	0.02
Sit and reach, cm	−3.0	(−4.7, −1.2)	0.4	(−2.2, 3.0)	0.047	0.77
Five chair sit-to-stands, s	−0.7	(−2.1, 0.7)	−1.2	(−3.1, 0.8)	0.71	0.27
Timed up and go, s	0.5	(−0.9, 1.8)	−1.6	(−3.4, 0.4)	0.10	0.70
Usual gait speed, s	1.0	(−0.2, 2.2)	0.1	(−1.6, 1.7)	0.39	0.71
Hand working with peg board, n	1.4	(−0.1, 2.8)	1.2	(−0.7, 3.1)	0.90	0.08

Data are presented as mean changes (95% confidence interval). * Adjusted for age, sex, educational attainment, and the corresponding baseline value of cognitive function tests. ** Adjusted for age, sex, and the corresponding baseline value of physical function tests.

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
