# Peer review of "A Novel Exercise for Enhancing Visuospatial Ability in Older Adults with Frailty: Development, Feasibility, and Effectiveness"

_geriatrics, 2020, doi:10.3390/geriatrics5020029_

Round 1

Reviewer 1 Report

The authors present a non-randomized feasibility study into a visuospatial and physical exercise program they developed, called the “cube exercise.” This study mostly does a good job of not presenting itself more than what it is: a feasibility study. At times they do go into the efficacy too much for a nonrandomized trial, and I’d caution how much time they spend on that aspect in general in the manuscript.

Some specific critiques:

Please clarify right from the beginning that both interventions were physical exercise interventions. Using the word “exercise” without describing it as either physical or cognitive (or both) was confusing for the first part of the article.

Participants were excluded with neurologic disease. Were they excluded for dementia? Were all subjects recruited with frailty cognitively normal at baseline?

I disagree that the intervention as described includes visual memory. It appears that the image they were to copy was in front of them to work with as they worked. Referring to any of this as memory would thus be misleading. Rather it is visuospatial/visuoconstruction skills and attention/concentration.

In a feasibility study, I think it would be important to note why there are so fewer subjects recruited into the cube group than the locomotive group. Please provide data on how many were approached, how many agreed to participate, and reasons provided not to participate (if available). Thus, for feasibility, are people interested in this type of intervention? Why or why not? Retention is covered well.

What is the rationale for looking both at overall scoring for the clock drawing (which was not significant) and qualitative change scores (which ended up being significant) in the trial? Were improved scores only within group, or were between group differences examined? I would be concerned that the qualitative analysis could not be blinded in this non-randomized trial and thus biased. I would also be concerned that qualitative analysis was not decided upon apriori, but rather added after the other scoring method was not significant. I would be tempted to cut the qualitative analysis unless the authors can supply a good rationale for this.

It does not appear that scores were adjusted for multiple comparisons. It is likely that sit-to-stand would no longer be significant. As the authors are likely underpowered, I would encourage them to add effect sizes to each of their outcomes and not just significance level.

How do the authors account for the fact that the cube intervention group was slower, weighed more, and had higher BMI at baseline, thus potentially allowing them more room for improvement after a physical exercise intervention?

Reviewer 2 Report

Review of geriatrics-743559 “A Novel Exercise for Enhancing Visuospatial Ability in Older Adults with Frailty: Development, Feasibility, and Effectiveness”

In this manuscript, the authors describe a cognitive training exercise that they label “cube exercise”. They hope that their newly-developed exercise may have advantages in “simultaneously enhancing both intellectual activity and physical activity”--two elements "that could be preventive factors of frailty".

I felt that this manuscript appears to be suitable for this journal. In the following, I provide further comments and suggestions:

(1) Table 1 and 2: the p values alone are not informative, because they are sample-size dependent; please add appropriate measures of effect size everywhere (e.g., Cohen’s d, R or R square, etc.)

(2) Abstract: the training task of the locomotive condition should be briefly described/mentioned in the Abstract, too.

(3) P.3 exclusion criteria: “being judged as ineligible by the principal investigator”. This is too vague. What were the criteria to be judged as ineligigble (i.e., please provide examples)?

(4) Figure 1/cube exercise: It seems that level 1 is two-dimensional and levels 2 and 3 both three-dimensional? Please clarify.

(5) Discussion: the small sample size (i.e., relatively low statistical power of the study) and the non-random assignment to groups must be critically discussed as a potential limitation of the study in the Discussion section!

Further points:

(6) Typography: figure 1 caption and elsewhere “dementional” should read “dimensional”

(7) P.3 “The other inclusion criteria were: individuals aged 65 or older, and able to walk with or without a walking aid.” The second inclusion criterion does not make sense, because it includes both possible events: “with or without aid”. This should be clarified or removed.

(8) “Table 2 shows the 12-week between-group changes in cognitive and physical functions” This sentence should be changed to: “Table 2 shows the 12-week within-group changes in cognitive and physical functions for the two groups.”
